# The Recent Decline of Apalachicola–Chattahoochee–Flint (ACF) River Basin Streamflow

Bin Fang [1,*], Jonghun Kam [2], Emily Elliott [3], Glenn Tootle [4], Matthew Therrell [3] and Venkat Lakshmi [1]

1    Engineering Systems and Environment, University of Virginia, Charlottesville, VA 22904, USA
2    Division of Environmental Science and Engineering, Pohang University of Science and
     Technology (POSTECH), Pohang 37673, Korea
3    Department of Geography, The University of Alabama (UA), Tuscaloosa, AL 35487, USA
4    Department of Civil, Construction, and Environmental Engineering, The University of Alabama (UA),
     Tuscaloosa, AL 35487, USA
*    Correspondence: bf3fh@virginia.edu

**Abstract:** The Apalachicola–Chattahoochee–Flint (ACF) basin is arguably the most litigated interstate river system in the eastern United States. Given the complicated demands for water use within this basin, it has been difficult to ascertain if the recent multi-decadal decline in streamflow is a product of human disturbance, changing climate, natural variability, or some combination of the above factors. To overcome these challenges, we examined unimpaired streamflow and precipitation within and adjacent to the ACF basin, upstream of the Apalachicola River at Chattahoochee, and the Florida streamflow station (ARCF), which has historically been identified to be representative of hydrologic variability in the ACF basin. Several of the upstream, unimpaired, streamflow stations selected were identified in rural watersheds where land-cover changes and human disturbance were minimal during the study period. When applying a series of statistical evaluations, ARCF streamflow variability generally reflects the natural variability of the ACF basin. Additionally, unimpaired streamflow variability from the neighboring Choctawhatchee River compared favorably with ARCF variability. The recent multi-decadal decline was consistent in all records, with the 2000s being the most severe in the historic record.

**Keywords:** streamflow decline; hydrologic variability; precipitation temporal variation





## 1. Introduction

The interstate waters that flow through the Apalachicola–Chattahoochee–Flint (ACF) river basin provide a crucial resource to the southeastern United States. The ACF basin contributes to water demands of metropolitan Atlanta, instream flow for agriculture in southwest Georgia, energy production in Alabama, and the essential freshwater needed for Florida's shellfish industry. A combination of multiple droughts over the past three decades in the southeastern U.S., along with immense population growth in the region, has resulted in a strain of this much needed and once abundant resource—water—resulting in litigation around sharing ACF waters. While conflict around water resources has been contested in the western U.S. (aka the Colorado River) for over a century, the southeastern U.S. has been largely isolated from this problem due to abundant instream flow availability. However, the Tri-State Water Wars between Alabama, Florida, and Georgia over the waters of the ACF have been waged in U.S. courts since 1990, reaching the U.S. Supreme Court on multiple occasions.

With the increased demands for water within the basin, as well as the multi-decadal trends in declining streamflow throughout the region, it has been difficult to parse the complicated interactions between human disturbance, changing climate, and natural hydroclimatic variability within the region. Large-scale climate forcing mechanisms, such as the high-frequency El Nino Southern Oscillation (ENSO) and the low frequency Atlantic

Multidecadal Oscillation (AMO), have long been studied and associated with hydrologic response in the southeastern U.S. [1–10]. The consensus of several of these studies confirms that El Nino (La Nina) is associated with increased (decreased) moisture, and that the AMO cold phase (warm phase) is associated with increased (decreased) moisture. Furthermore, several studies utilized Pacific and Atlantic Ocean Sea Surface Temperatures (SSTs) to verify and confirm these climate signals in southeastern U.S. streamflow [11,12]. While a plausible cause of the recent southeastern U.S. streamflow decline may be explained by a combination of a persistent AMO Warm phase and multiple La Nina events in the 2000s, the question arises: has the behavior of the ACF basin been consistent with other regional southeastern U.S. basins? Specifically, are the significant declines in the interstate ACF river system in tune with the surrounding watersheds, or do the declines in streamflow in the ACF indicate further complicating influences in the system, such as local/regional human disturbance? While it is widely acknowledged that human disturbance influences streamflow data collected at the USGS station, the Apalachicola River at Chattahoochee, Florida (ARCF), the degree or significance of this disturbance is widely debated. Thus, the motivation of the current research was to determine if the historic variability of ARCF station streamflow, including the recent multi-decadal decline, is consistent with the natural variability in the ACF watershed/sub-basins and adjacent watersheds. Previously, several studies applied various machine learning methodologies or GIS tool to simulate or predict streamflow discharges by using historical data [13–15].

## 2. Materials and Methods

Streamflow data from nine gauge stations were obtained from the U.S. Geological Survey (USGS) National Water Information System (NWIS) (Figure 1, Table 1). The nine gauge stations include seven streamflow stations within the ACF basin (hereby referred to as ACF sub-basin stations), the Choctawhatchee River near Bruce, Florida (CRBF) streamflow station, and the ARCF streamflow station. The monthly streamflow data were retrieved from the National Water Information System (NWIS) website (http://waterdata.usgs.gov/nwis) (accessed on 1 July 2022) in cubic feet per second (cfs) for each month in the calendar year and were converted into streamflow volumes (Million Cubic Meters—MCM) using the appropriate conversion factors. The NWIS streamflow measurements were collected by acoustic Doppler current profilers (ADCP). The validation experiments showed that the mean differences between tow cart velocity and ADCP bottom track and water track velocities were −0.51 and −1.10%, respectively [16,17]. Using the months in the calendar year, the cumulative annual streamflow volume was used in the current research, with periods of record from 1961 to 2017 (57 years) and 1942 to 2017 (76 years), given that complete records of data vary for the streamflow stations selected. For this study, eight unimpaired streamflow stations (stations identified with minimal anthropogenic influences) were selected [18,19] to isolate the impacts of significant anthropogenic influence in the analysis. These unimpaired streamflow stations included the seven ACF sub-basin stations and the CRBF station. The streamflow records for all stations were complete except for two years, 1983 and 1984, for the CRBF station. We used the cumulative annual streamflow volumes for 1983 and 1984 from an upstream, unimpaired station (Choctawhatchee River at Newton, Alabama; USGS 02361000) and interpolated (linearly) based on comparison of the annual flows at both stations.

This study includes four statistical evaluations, including (1) Correlation (Statistical Significance—%) and Coefficient of Determination ($R^2$) (2) Stability—$R^2$ average and $R^2$ minimum (3) Linear Trend Analysis—positive or negative trend and trend slope (%) (4) Multiple Linear Regression (Forwards and Backwards Stepwise)—Model Coefficients.

(1) The annual ARCF streamflow was correlated with annual streamflow from the seven ACF sub-basin stations and annual streamflow from the CRBF streamflow station. Statistical significance (percentage) and the Coefficient of Determination ($R^2$) were determined for the entire period of record (1961 to 2017 or 1942 to 2017).

(2) While an evaluation of the entire period of record is effective for determining the strength of the linear relationship for such period, the possibility exists that periods of poor correlation could reside within this entire period of record. Thus, to evaluate this, Stability Analysis was performed, such that a 10-year moving correlation window was used to evaluate the correlation of the annual ARCF streamflow with the annual streamflow from the seven ACF sub-basin stations and the CRBF streamflow station.

(3) Annual streamflow for all stations was standardized (zero mean and standard deviation of one) and a Linear Trend Analysis was performed. This involved the fitting of a trend line to the annual streamflow for the appropriate period of record (1961 to 2017 or 1942 to 2017). The fitted line will reveal (1) if the trend is positive or negative and (2) the slope (percentage—%) of the trend line. The slope of the trend line reveals the rate of either increase (or decrease) in annual streamflow over time. The standardization of the data allows for a consistent comparison of trends across all basins (ARCF, ACF sub-basin stations, CRBF).

(4) Multiple Linear Regression, applying a Forward and Backwards Stepwise model, was applied, and the independent variable (ARCF annual streamflow) was regressed against the dependent variables (ACF sub-basin streamflow). Standardized annual streamflow was used in this analysis. A standard stepwise regression adds and removes predictors (e.g., ACF sub-basin streamflow), as needed, for each step. The model stops when all variables not in the model have $p$-values that are greater than the specified alpha-to-enter value and when all variables in the model have $p$-values that are less than or equal to the specified alpha-to-remove value. The F-level for a predictor had to have a maximum $p$-value of 0.05 for entry and 0.10 for retention in our stepwise regression model. The motivation of developing the regression model was to focus on the coefficients (i.e., weights) developed for the regression equation. Given all the data were standardized, the coefficients reflected the strength (or weakness) of the dependent variable, and thus the relationship of the dependent variable(s) (annual ACF sub-basin streamflow) with the independent variable (ARCF annual streamflow).

**Table 1.** Information of the 9 USGS stations: ID, station name and identifier, USGS station number (#), period of record, watershed area, latitude, and longitude.

| ID | USGS Station Name and Identifier | USGS Station # | Period of Record | Watershed Area (km$^2$) | Latitude | Longitude |
|---|---|---|---|---|---|---|
| 1S | Uchee Creek near Fort Mitchell, AL (UCH) | 02342500 | 1961–2017 | 834 | 32.32 | −85.01 |
| 2S | Turkey Creek at Byromville, GA (TUR) | 02349900 | 1961–2017 | 124 | 32.20 | −83.90 |
| 3S | Snake Creek near Whitesburg, GA (SNK) | 02337500 | 1961–2017 | 93 | 33.53 | −84.93 |
| 4S | Flint River at US 19, near Carsonville, GA (FLT) | 02347500 | 1942–2017 | 4791 | 32.72 | −84.23 |
| 5S | Chestatee River at St Rt 52 near Dahlonega, GA (CHR) | 02333500 | 1942–2017 | 396 | 34.53 | −83.94 |
| 6S | Sweetwater Creek near Austell, GA (SWT) | 02337000 | 1942–2017 | 616 | 33.77 | −84.61 |
| 7S | Ichawaynochaway Creek at Milford, GA (ICH) | 02353500 | 1942–2017 | 1606 | 31.38 | −84.55 |
| 8S | Choctawhatchee River near Bruce, FL (CRBF) | 02366500 | 1942–2017 | 11355 | 30.55 | −85.90 |
| 9S | Apalachicola River at Chattahoochee, FL (ARCF) | 02358000 | 1961–2017 & 1942–2017 | 44548 | 30.70 | −84.86 |

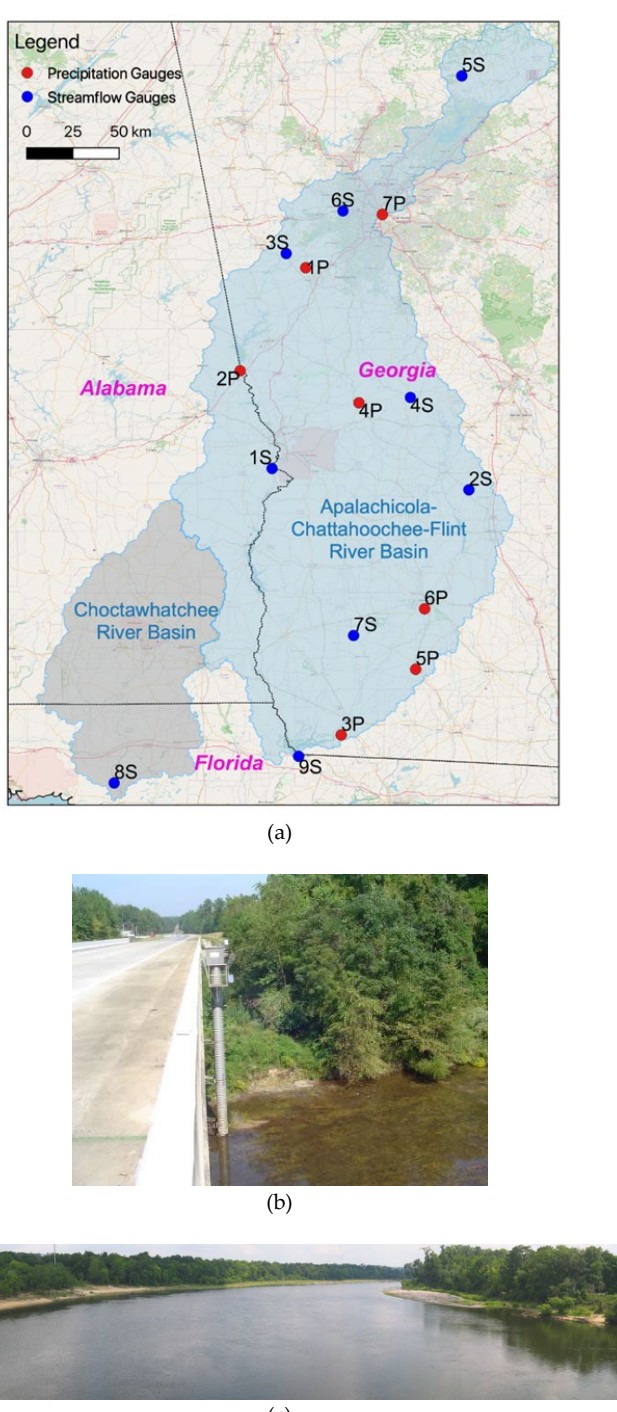

**Figure 1.** (**a**) Location Map showing ACF and Choctawhatchee basins with streamflow and COOP station locations. The ACF basin is in light blue and the Choctawhatchee basin is in light gray. Streamflow stations are blue dots: 1S (Uchee Creek), 2S (Turkey Creek), 3S (Snake Creek), 4S (Flint River), 5S (Chestatee River), 6S (Sweetwater Creek), 7S (Ichawaynochaway Creek), 8S (Choctawhatchee River near Bruce, and 9S (Apalachicola River at Chattahoochee, Florida). The NOAA Rainfall COOP stations are identified as red dots: 1P (Newnan GA), 2P (West Point GA), 3P (Bainbridge GA), 4P (Talbotton GA), 5P (Camilla GA), 6P (Albany GA), and 7P (Atlanta GA). Two photographs show surroundings near USGS stations 1S and 8S (**b**) Photo of Uchee Creek near Fort Mitchell, AL (1S) (Image source: https://waterdata.usgs.gov/nwis/ accessed on 20 June 2022) (**c**) Photo of Apalachicola River at Chattahoochee, FL (8S) (Image source: https://waterdata.usgs.gov/nwis/ accessed on 20 June 2022).

## 3. Results

When correlating ARCF streamflow with each of the eight streamflow stations (seven ACF sub-basin stations and the CRBF streamflow station) for the entire period of record (either 1961 to 2017 or 1942 to 2017), statistical significance exceeded 99% for all streamflow stations, thus revealing that natural variability is a key driver of annual streamflow statistics at the ARCF station. Coefficient of Determination ($R^2$) values ranged from a low value of $R^2$ = 0.55 for the Chestatee River (CHR) to a high value of $R^2$ = 0.88 for Ichawaynochaway Creek (ICH) (Table 2).

**Table 2.** Statistical variables of the 9 USGS stations, including: $R^2$, $R^2$ Stability (Average), $R^2$ Stability (Minimum), Trend Slope (Positive or Negative), Trend Slope (%), Model Coefficients, and Cumulative Deficit Streamflow (1999–2017) measured in Standard Deviations.

| USGS Station Name | Period of Record | $R^2$ | $R^2$ Stability (Average) | $R^2$ Stability (Minimum) | Trend Slope | Trend Slope | Model Coefficients | Deficit Streamflow (1999–2017) |
|---|---|---|---|---|---|---|---|---|
| Uchee Creek (UCH) | 1961–2017 | 0.77 | 0.68 | 0.05 | Negative | −1.5% | 0.32 | −8.7 |
| Turkey Creek (TUR) | 1961–2017 | 0.79 | 0.81 | 0.62 | Negative | −1.9% | 0.45 | −10.3 |
| Snake Creek (SNK) | 1961–2017 | 0.69 | 0.63 | 0.04 | Negative | −3.3% | 0.28 | −16.0 |
| Flint River (FLT) | 1942–2017 | 0.82 | 0.83 | 0.60 | Negative | −1.4% | 0.29/0.27 | −12.3 |
| Chestatee River (CHR) | 1942–2017 | 0.55 | 0.52 | 0.00 | Negative | −0.9% | 0.11/0.10 | −11.5 |
| Sweetwater Creek (SWT) | 1942–2017 | 0.62 | 0.65 | 0.01 | Negative | −0.1% | 0.10/0.10 | −6.6 |
| Ichawaynochaway Creek (ICH) | 1942–2017 | 0.88 | 0.88 | 0.67 | Negative | −1.2% | 0.56/0.43 | −11.3 |
| Choctawhatchee River (CRBF) | 1942–2017 | 0.82 | 0.84 | 0.53 | Negative | −0.6% | NA/0.18 | −7.9 |
| Apalachicola River (ARCF) | 1961–2017 & 1942–2017 | NA | NA | NA | Negative | −2.2%/−1.1% | NA | −11.6 |

When correlating ARCF streamflow with each of the eight (seven ACF sub-basins and the CRBF streamflow station) streamflow stations for 10-year periods within the entire period of record (either 1961 to 2017 or 1942 to 2017), referred to as Stability Analysis, average $R^2$ stability values ranged from 0.52 to 0.88. For example, for the 1961 to 2017 period of record, the first 10-year period was 1961 to 1970, and thus, an $R^2$ stability value was determined when correlating ARCF streamflow and the streamflow station of interest. The next 10-year window was 1962 to 1971 and, again, an $R^2$ stability value was determined. This was repeated and, for the 1961 to 2017 period of record, forty-eight $R^2$ stability values were determined, while for the period from 1942 to 2017, sixty-seven $R^2$ stability values were determined. The average $R^2$ stability values were calculated by averaging the forty-eight (or sixty-seven) $R^2$ stability values for each streamflow station. Perhaps of greater interest is examination of the minimum $R^2$ stability value within the vector of forty-eight (or sixty-seven) $R^2$ stability values, as this could reveal potential challenges in the relationship between ARCF streamflow and streamflow from the station of interest. Referring to Table 2, for the 1961 to 2017 period of record, the Uchee Creek (UCH) and Snake Creek (SNK) stations showed the minimum $R^2$ stability values close to zero. A further visual analysis of the forty-eight $R^2$ stability values for each streamflow station revealed a consistent pattern of acceptable $R^2$ stability values, except for one 10-year period from 1989 to 1998, whereas the minimum $R^2$ stability value identified occurred for both streamflow stations. The

remaining station (Turkey Creek (TUR)) evaluated during the 1961 to 2017 period of record displayed both overall and consistent stability. Referring again to Table 2, for the 1942 to 2017 period of record, the CHR and Sweetwater Creek (SWT) stations had minimum $R^2$ stability values close to zero. A further visual analysis of the sixty-seven $R^2$ stability values for each streamflow station revealed a consistent pattern of acceptable $R^2$ stability values, except for one 10-year period from 1989 to 1998, whereas the minimum $R^2$ stability value identified occurred for both streamflow stations. Thus, four streamflow stations (UCH, SNK, CHR, and SWT) with locations varying spatially across the ACF basin identified a consistent period (1989 to 1998) in which sub-basin streamflow was poorly correlated with ARCF streamflow, and this period appears to be the only period in the record that displays such poor correlation. The remaining stations evaluated during the 1942 to 2017 period of record—the Flint River (FLT), ICH, and CRBF stations—displayed both overall and consistent stability.

Linear Trend Analysis involved the fit of a line to the time series of a standardized annual streamflow for all streamflow stations, including the ARCF station. This resulted in determining a regression equation for each streamflow station and the slope (positive or negative) of the line associated with such regression equation. The use of standardized annual streamflow allows for comparison, given the varying magnitude differences in annual streamflow due to varying watershed areas. All nine streamflow stations revealed a negative trend, with slopes ranging from −0.1% to −3.3%. For the 1961 to 2017 period of record, the ARCF slope was −2.2%, while for the 1942 to 2017 period of record, the ARCF slope was −1.1%. The increase in the downward trend when comparing the more recent (1961 to 2017) period of record to the 1942 to 2017 period of record was consistent in the streamflow stations. This supports the recent decline in streamflow identified throughout the southeast United States [12].

Regression models were developed using standardized annual streamflow for each (1961 to 2017 or 1942 to 2017) period of record. For the 1961 to 2017 period of record, a model was developed with ARCF streamflow (dependent variable) and three predictors (independent variables), UCH, TUR, and SNK. Model coefficients revealed relatively similar weights (0.32, 0.45, 0.28, respectively), and thus, each streamflow station generally contributed equally to the regression model (Table 2). For the 1942 to 2017 period of record, a model was developed with ARCF streamflow (dependent variable) and four predictors (independent variables), including the FLT, CHR, SWT, and ICH stations. Model coefficients revealed the ICH and FLT stations contributed the greatest to the regression model (0.56 and 0.29, respectively) (Table 2). An additional regression model was developed for the 1942 to 2017 period, in which the CRBF was added as an independent variable. The ICH (0.43) and FLT (0.27) were again the greatest contributors to the regression model, while the CRBF (0.18) was ranked third, ahead of the CHR (0.10) and SWT (0.10) (Table 2).

## 4. Discussion and Conclusions

While some variation occurred across the statistics (e.g., $R^2$, $R^2$ Stability Average, $R^2$ Stability Minimum, Trend Slope, and Model Coefficient) evaluated in this study, there was a consensus of similar behavior across the ACF basin and the neighboring CRBF station. ARCF streamflow generally displayed similar temporal variability when compared to eight unimpaired streamflow stations, and thus, ARCF streamflow appears to reflect the natural variability of the ACF basin and the neighboring CRBF station. A 10-year end-year filter was applied to the annual standardized streamflow for each streamflow station for each period of record (1961 to 2017 or 1942 to 2017) and, visually, the temporal variability of ARCF streamflow compared very favorably with the ACF sub-basin streamflow and the CRBF station, thus supporting this statistical evaluation (Figure 2a,b). The 10-year filtered lows identified in the 2000s were generally the lowest in the historic record, although a slight shift towards a positive trend was observed near the end of the period of record (~2010 to 2017) (Figure 2a,b). A recent study of southeast U.S. streamflow, in which 26 unimpaired streamflow stations were examined for the period of 1952 to 2016, displayed

similar behavior when compared to the ACF basin [12]. It is worth noting that no ACF streamflow stations were included in the [12] study. Thus, the current study confirmed that the behavior (temporal variability) of the impaired ARCF streamflow station and the unimpaired ACF sub-basin streamflow stations, and, the recent multi-decadal decline, compares very favorably with streamflow variability across the southeast United States.

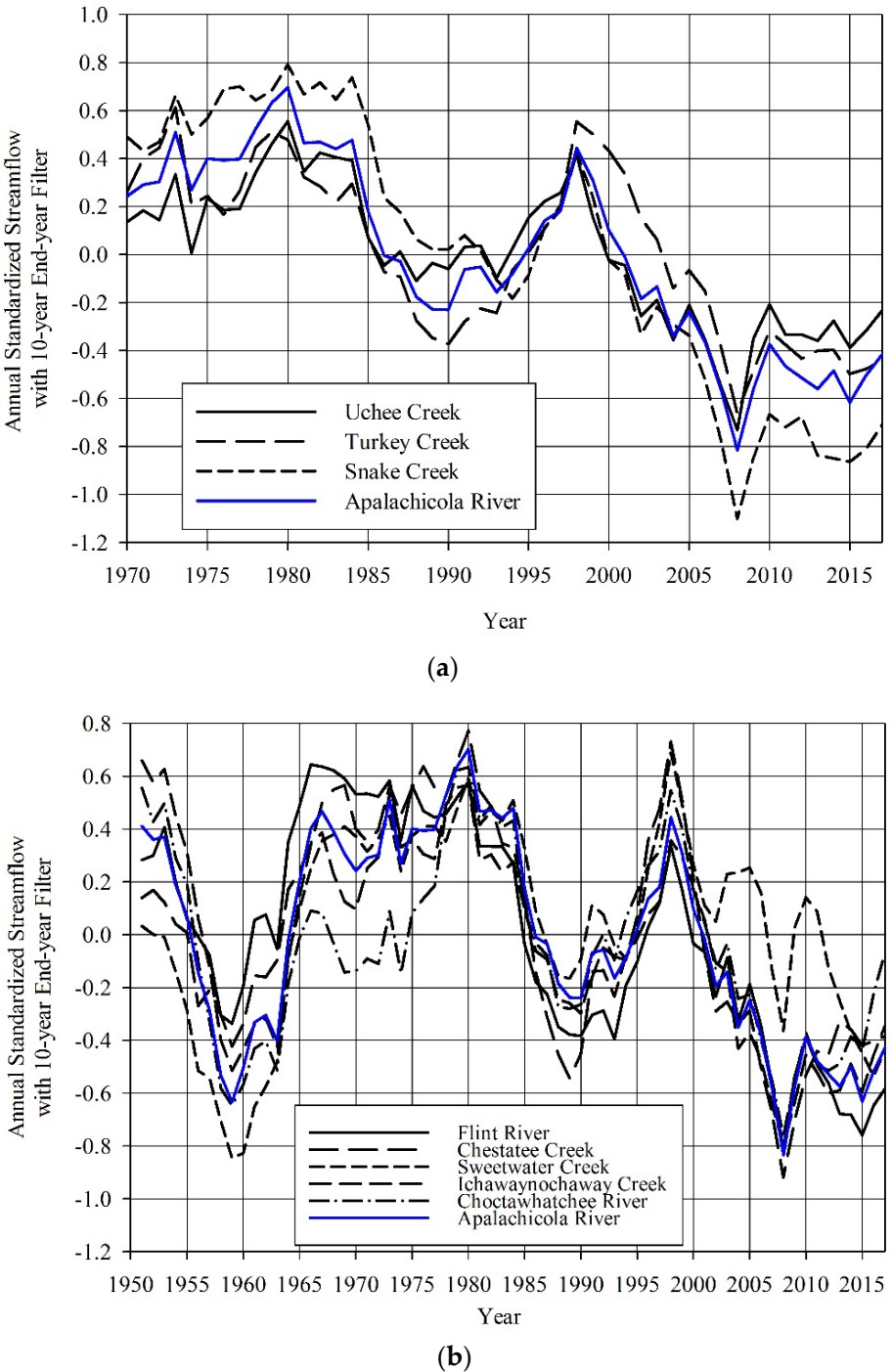

**Figure 2.** Annual standardized streamflow with 10-year end-year filter (**a**) Streamflow: 1970–2017 (based on annual data of 1961–2017) (**b**) streamflow: 1951–2017 (based on annual data of 1942–2017).

Perhaps the one exception in the various statistics evaluated was the minimum $R^2$ stability values for four stations (UCH, SNK, CHR and SWT) in which the period of 1989 to 1998 was found to display anomalously low $R^2$ stability values. The 1997/1998

El Nino was perhaps the strongest on record and was associated with above average moisture throughout the southeast United States. For the 1961 to 2017 period of record, the UCH, TUR, SNK, and ARCF stations' 1998 annual streamflow (average of the four stations) was +1.34 standard deviations above average. The following year (1999), these same four streamflow stations had an average annual flow of −1.31 standard deviations below average. From 1999 to 2017, cumulative streamflow deficits for the UCH, TUR, and SNK stations ranged from −8.7 to −16.0 standard deviations (Table 2). Thus, for the nineteen-year period (1999 to 2017), average annual streamflow deficits ranged from −0.5 to −0.8 standard deviations per year. For the 1942 to 2017 period, the FLT, CHR, SWT, ICH, CRBF, and ARCF station's 1998 annual streamflow (average of the six stations) was +1.18 standard deviations above average. The following year (1999), these same six streamflow stations had an average annual flow of −1.30 standard deviations below average. The shift from increasing streamflow in the 1990s to a period of declining streamflow beginning in the late 1990s is clearly visible, along with the 1998 peak (Figure 2a,b). From 1999 to 2017, cumulative streamflow deficits for these six stations ranged from −6.6 to −12.3 standard deviations (Table 2). Thus, for the nineteen-year period (1999 to 2017), average annual streamflow deficits ranged from −0.3 to −0.6 standard deviations per year for the six stations. The 1999 to 2017 declines in streamflow were statistically significant. When applying the rank-sum test [10,20] to evaluate if there was a statistical difference in 1942 to 1998 (or 1961 to 1998) streamflow in comparison with the 1999 to 2017 streamflow, ARCF streamflow significance exceeded 99%, as did streamflow from six of the seven ACF sub-basin stations. The lone exception, SWT streamflow, did exceed 95% significance. The nineteen-year (1999 to 2017) period was impacted by the strengthening of the warm phase of the Atlantic Multi-decadal Oscillation (AMO), which began around 1995, and multiple La Nina events in the 2000s. Each of these climate phenomena are associated with reduced moisture (streamflow and precipitation) in the southeast U.S. [7,10]. The coupling of AMO Warm and La Nina displayed significantly lower annual (water year) precipitation in the ACF basin [10]. Thus, a possible explanation of the behavior of the 1990s (1989–1998) and poor stability results may be associated with the transition of the AMO from a cold to a warm phase.

Annual precipitation (inches converted to mm) from 1942 to 2017 was obtained for seven stations (Newnan GA–1P (33.45, −84.82), West Point GA–2P (32.87, −85.19), Bainbridge GA–3P (30.82, −84.62), Talbotton GA–4P (32.69, −84.52), Camilla GA–5P (31.19, −84.20), Albany GA–6P (31.53, −84.15) and Atlanta GA–7P (33.63, −84.44)) within the ACF basin from the NOAA Cooperative Observer Network (COOP) website (https://www.ncdc.noaa.gov/data-access/land-based-station-data/land-based-datasets/cooperative-observer-network-coop) (accessed on 1 July 2022) (Figure 1). A 10-year end-year filter was applied to annual standardized precipitation for each station, and the temporal variability of the precipitation stations was visually examined (Figure 3a). The temporal variability of the precipitation stations compares favorably to streamflow variability in the ACF. The recent decline in both streamflow and precipitation is perhaps the most severe in the historic record. It appears the declining trend may be shifting to a positive trend, as reflected by both streamflow and precipitation. This positive trend may be attributed to the possibility of the AMO moving from a warm to cold phase.

For the 10-year end-year filter, we compared the average precipitation for the seven NOAA COOP stations (Figure 3a—red line) to the ARCF streamflow (Figure 2b—blue line) in Figure 3b. The Coefficient of Determination ($R^2$) value was 0.80, revealing the strong (and expected) relationship with the average annual precipitation for the seven NOAA COOP stations (hereby referred to as ACF basin precipitation) and ARCF streamflow. While Figure 3b reveals a divergence between ACF basin precipitation and ARCF streamflow in the ~2000s (drought period), in that ARCF streamflow underperforms when compared to ACF basin precipitation, this behavior was not unprecedented. The late ~1950s drought (aka second Dust Bowl drought period) displayed similar behavior (Figure 3b).

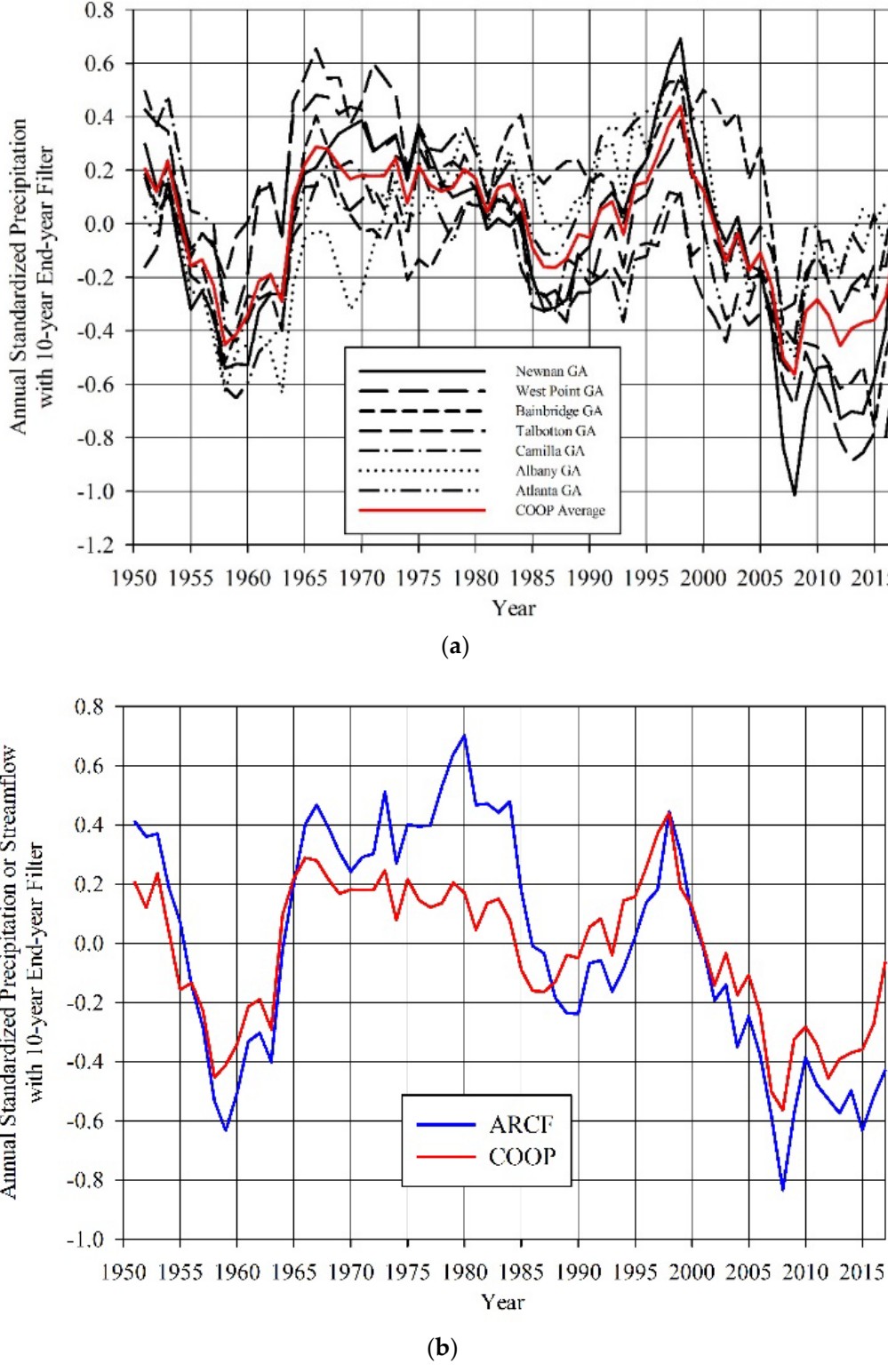

**Figure 3.** Annual standardized precipitation or streamflow with 10-year end-year filter from 1951–2017 (based on annual data of 1942–2017) (**a**) NOAA COOP precipitation stations (Newnan GA, West Point GA, Bainbridge GA, Talbotton GA, Camilla GA, Albany GA, Atlanta GA, and COOP Average (**b**) COOP Average and ARCF streamflow.

In evaluating a basic water balance model (Inflow − Outflow = Delta Storage) of an unimpaired watershed, Precipitation (P) and Groundwater-In are the major influxes while Runoff (i.e., streamflow) (Q), Groundwater-Out, and Evapotranspiration (ET) are the main

outfluxes. We assume (1) net change in watershed storage over a year is negligible; thus, Delta Storage is zero; (2) Groundwater-In is nearly equal to Groundwater-Out for a large watershed. Thus, the water balance equation reduces to ET = P − Q. This approach is commonly referred to as the Bucket Model [21,22]. Applying this approach, ET would represent the net losses in an unimpaired watershed. However, in an impaired watershed, net losses would include ET, diversions, withdrawals, and accumulations of water due to impoundments. While streamflow (Q or runoff) and precipitation (P) revealed similar temporal variability, assessing net losses for the ARCF streamflow station and comparing such net losses to ACF sub-basins stations may reveal the impacts of human disturbance. For the 1942 to 2017 period of record, the annual streamflow volume (MCM) was converted to centimeters (cm) by dividing by the watershed area (km$^2$) with proper conversions for the ARCF, FLT, SWT, and ICH streamflow stations. Annual precipitation (cm) for each streamflow station was determined by assigning a specific precipitation station or stations based on their spatial proximity to the streamflow station's watershed (Figure 1): ARCF (9S)—average of all seven precipitation stations; FLT (4S)—Talbotton (4P); SWT (6S)—Atlanta (7P); ICH (7S)—average of Camilla (5P) and Albany (6P). For the 76-year period of record (1942 to 2017), the average annual net loss [P (cm)–Q (cm)] for each station was: ARCF (84.7 cm), FLT (89.5 cm), SWT (76.8 cm), and ICH (86.6 cm). Thus, the average annual net loss for the three ACF sub-basin stations was 84.3 cm, confirming that net losses for the ARCF station were similar to net losses in the unimpaired ACF sub-basins. Note that the CHR (5S) station was not evaluated, as it is spatially located in the northern ACF basin with no nearby precipitation station.

The Southeastern U.S. has experienced tremendous growth, from metropolitan Atlanta to the State of Florida, whose population will likely approach 26 million residents per the projected 2030 U.S. Census. Water resources are being modified (human disturbance) due to society's dependence on water. The current research, studying the relationship between streamflow and precipitation, allows us to better understand these impacts. The current research examined the ACF basin and provides critical insights as to the recent decline by displaying a similar relationship between precipitation and streamflow over the last ~75 years. The current research revealed that the ARCF station does reflect the natural variability of the watershed, as this station displayed a similar temporal pattern to upstream unimpaired stations, precipitation, and hydrologic variability of the neighboring Choctawhatchee River basin. Thus, the contributions of this study should aid in the ongoing discussions of sharing water between the states of Alabama, Georgia, and Florida. The current research focuses only on two river basins in the Southeast US and uses streamflow data from nine USGS gauges. In the future, we will extend our research to other regions of different climate types and terrain and land cover conditions in the US.

**Author Contributions:** Conceptualization, G.T. and B.F.; methodology, G.T. and J.K.; software, B.F.; validation, B.F. and J.K.; formal analysis, G.T. and B.F.; investigation, G.T.; resources, V.L. and M.T.; data curation, G.T. and B.F.; writing—original draft preparation, G.T., E.E. and M.T.; writing—review and editing, E.E. and V.L.; visualization, J.K. and G.T.; supervision, G.T., V.L. and M.T.; project administration, G.T.; funding acquisition, G.T. and M.T. All authors have read and agreed to the published version of the manuscript.

**Funding:** Support provided by the U.S. Environmental Protection Agency Gulf of Mexico Program (EPA-MX-00D67718-0) and the National Science Foundation Paleo Perspectives on Climate Change (18059590).

**Institutional Review Board Statement:** Not applicable.

**Informed Consent Statement:** Not applicable.

**Data Availability Statement:** The data presented in this study are openly available in "Research data and codes for GRL paper", Fang, Bin, 2020, University of Virginia Dataverse, V1, https://doi.org/10.18130/V3/4MMOLV accessed on 20 June 2022.

**Acknowledgments:** Support provided by the U.S. Environmental Protection Agency Gulf of Mexico Program (EPA-MX-00D67718-0) and the National Science Foundation Paleo Perspectives on Climate Change (18059590).

**Conflicts of Interest:** The authors declare no conflict of interest.

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
