# Peer review of "The Recent Decline of Apalachicola–Chattahoochee–Flint (ACF) River Basin Streamflow"

_hydrology, doi:10.3390/hydrology9080140_

Round 1
Reviewer 1 Report
1. The manuscript presents the recent decline of Apalachicola-Chattahoochee-Flint river basin streamflow, which is interesting. The subject addressed is within the scope of the journal.
2. However, the manuscript, in its present form, contains several weaknesses. Appropriate revisions to the following points should be undertaken in order to justify recommendation for publication.
3. For readers to quickly catch your contribution, it would be better to highlight major difficulties and challenges, and your original achievements to overcome them, in a clearer way in abstract and introduction.
4. It is shown in the reference list that the authors have several publications in this field. This raises some concerns regarding the potential overlap with their previous works. The authors should explicitly state the novel contribution of this work, the similarities, and the differences of this work with their previous publications.
5. p.1 - Apalachicola-Chattahoochee-Flint river basin is adopted as the case study. What are other feasible alternatives? What are the advantages of adopting this case study over others in this case? How will this affect the results? The authors should provide more details on this.
6. p.2 - historical records of 1942 to 2017 are taken. Why are more recent data not included in the study? Is there any difficulty in obtaining more recent data? Are there any changes to the situation in recent years? What are its effects on the result?
7. p.2 - the cumulative annual streamflow volume is adopted in the analysis. What are other feasible alternatives? What are the advantages of adopting this approach over others in this case? How will this affect the results? The authors should provide more details on this.
8. p.2 - eight unimpaired streamflow stations are adopted to isolate the impacts of significant anthropogenic influence in the analysis. What are the other feasible alternatives? What are the advantages of adopting these stations over others in this case? How will this affect the results? More details should be furnished.
9. p.3 - nine gauge stations as shown in Table 1 are adopted in the analysis. What are the other feasible alternatives? What are the advantages of adopting these gauge stations over others in this case? How will this affect the results? More details should be furnished.
10. p.4 - statistical significance (percentage) and the coefficient of determination are adopted to correlate Annual ARCF streamflow with annual streamflow from the seven ACF sub-basin stations and annual streamflow from the CRBF streamflow station. What are the other feasible alternatives? What are the advantages of adopting these tools over others in this case? How will this affect the results? More details should be furnished.
11. p.4 - stability analysis is adopted to evaluate the correlation. What are other feasible alternatives? What are the advantages of adopting this approach over others in this case? How will this affect the results? The authors should provide more details on this.
12. p.4 - a linear trend analysis is adopted to fit a trend line to the annual streamflow for the appropriate period of record. What are other feasible alternatives? What are the advantages of adopting this approach over others in this case? How will this affect the results? The authors should provide more details on this.
13. p.5 - multiple linear regression, applying a forward and backwards stepwise model, is adopted to regress ARCF annual streamflow against ACF sub-basin streamflow. What are other feasible alternatives? What are the advantages of adopting this approach over others in this case? How will this affect the results? The authors should provide more details on this.
14. p.6 - “…Perhaps the one exception in the various statistics evaluated was the minimum R2 stability values for four stations (UCH, SNK, CHR and SWT) in which the period of 1989 to 1998 was found to display anomalously low R2 stability values.…” Some justification should be furnished on this issue.
15. p.7 - “…a possible explanation of the behavior of the 1990’s (1989-1998) and poor stability results may be associated with the transition of the AMO from a Cold to Warm phase.…” More justification should be furnished on this issue.
16. Some key parameters are not mentioned. The rationale on the choice of the particular set of parameters should be explained with more details. Have the authors experimented with other sets of values? What are the sensitivities of these parameters on the results?
17. Some assumptions are stated in various sections. Justifications should be provided on these assumptions. Evaluation on how they will affect the results should be made.
18. The discussion section in the present form is relatively weak and should be strengthened with more details and justifications.
19. Moreover, the manuscript could be substantially improved by relying and citing more on recent literature about contemporary real-life case studies of soft computing techniques on hydrologic prediction and uncertainty such as the following. Discussions about result comparison and/or incorporation of those concepts in your works are encouraged:
● Fu, M.L., et al., “Deep Learning Data-Intelligence Model Based on Adjusted Forecasting Window Scale: Application in Daily Streamflow Simulation IEEE ACCESS 8: 32632-32651 2020.
● Taormina, R., et al., “ANN-based interval forecasting of streamflow discharges using the LUBE method and MOFIPS”, Engineering Applications of Artificial Intelligence 45: 429-440 2015.
● Kaya, C.M., et al., “Predicting flood plain inundation for natural channels having no upstream gauged stations,” Journal of Water and Climate Change 10 (2): 360-372 2019.
20. In the conclusion section, the limitations of this study, suggested improvements of this work and future directions should be highlighted.
Reviewer 2 Report
Please see the file attached.

Author Response
Attached includes response to reviewer 2.

Round 2
Reviewer 1 Report
The revised paper has addressed all my previous comments, and I suggest to ACCEPT the paper as it is now.
Reviewer 2 Report
I'm very satisfied with the Authors' replies to my concerns. I believe the manuscript can be accepted in present form. My congratulations to the Authors!